# Feelings of Personal Relative Deprivation and Subjective Well-Being in Japan

**DOI:** 10.3390/bs13020158

**Published:** 2023-02-11

**Authors:** Hiroshi Ohno, Kyung-Tae Lee, Takashi Maeno

**Affiliations:** 1Graduate School of System Design and Management, Keio University, Kyoseikan, 4-1-1 Hiyoshi, Kohoku-Ku, Yokohama 223-8526, Japan; 2Department of Marketing and International Trade, Faculty of Commerce, Chuo University, 742-1 Higashinakano Hachioji-shi, Tokyo 192-0393, Japan

**Keywords:** personal relative deprivation, subjective well-being, personal relative deprivation scale, social comparison orientation, Japan

## Abstract

Personal relative deprivation (PRD) refers to emotions of resentment and dissatisfaction caused by feeling deprived of a deserved outcome compared to some reference. While evidence suggests that relative deprivation based on objective data such as income affects well-being, subjective PRD has been less explored, especially in the East. This study evaluated the relationship between PRD and subjective well-being based on various aspects in the context of Japan. An online questionnaire survey, including the Japanese version of the Personal Relative Deprivation Scale (J-PRDS5) and various well-being indices, was administered to 500 adult participants, balanced for sex and age. Quantitative data analysis methods were used. PRD significantly correlated with subjective well-being as assessed by various aspects. Through mediation analysis, we found that a strong tendency to compare one’s abilities with others may undermine subjective well-being through PRD. The results also indicated that well-developed human environments may be associated with the maintenance of subjective well-being levels, even when PRD is high. Toward developing future interventions to improve well-being and health, efforts must be undertaken in Japan to monitor PRD and further clarify the mechanism of the association between PRD and the factors that showed a strong relationship in this study.

## 1. Introduction

According to the Gini index from 2017–2021 [1], Japan is highly ranked for its relatively large economic inequality among the Organisation for Economic Co-operation and Development (OECD) countries (13th out of 41 countries). Based on social survey data, Oishi et al. [2] (p. 1095) showed that Americans were “happier in the years with less national income inequality than in the years with more.” The same study suggested that the negative relationship between income inequality and the well-being of low-income individuals is not only explained by their low household income but also by a sense of inequality and a lack of well-being [2]. In Japan, the widening economic disparities might have increased feelings related to inequality and reduced the level of happiness among the Japanese people. The concept of personal relative deprivation (PRD) is an emotional and psychological factor related to a decline in happiness caused by persisting inequalities, such as economic disparity. Specifically, it refers to emotions of resentment and dissatisfaction caused by feeling deprived of a deserved outcome that one desires compared to some reference (e.g., what similar people have) [3,4]. 

The concept of relative deprivation was proposed by Stouffer et al. [5] in a social psychology study after World War II. The researchers argued that the extent to which one feels deprived and victimized depends largely on the standard used for comparison. The Gini index [6] is well-known as an approach to relative deprivation research based on objective income data. The study by D’Ambrosio and Frick [7] is also based on similar data, wherein the authors derived coefficients related to relative deprivation or satisfaction that was calculated from objective income data and found that relative deprivation/satisfaction was more closely related to happiness than the absolute amount of income. However, we believe that relative deprivation based on objective income data does not necessarily coincide with the subjective feeling of PRD. In recent years, the development of Personal Relative Deprivation Scale (PRDS) to study the feeling of relative deprivation perceived at the individual level [3,8,9,10] in Western countries has been observed. 

Several previous studies have shown that relative deprivation based on objective measures such as income data have a negative impact on subjective well-being [7,11]. Furthermore, some studies suggest that the subjectively assessed feeling of PRD is a significant predictor of mental health and depression [12,13]. Based on these previous studies, we hypothesize that a causal relationship exists between feelings of PRD and subjective well-being, as well as various indicators assessing multidimensional subjective well-being. However, limited research has empirically clarified this relationship; thus, the current study examines this hypothetical relationship and demonstrates the effects of PRD on various aspects of subjective well-being.

We aimed to elucidate the relationship between the feeling of PRD and subjective well-being in Japan. However, although the study on the Japanese translation of the PRDS [14], a measure of PRD, included 798 participants and clarified the reliability coefficient, scale structure, and correlation with a few indicators, the external criterion referenced-validity has not been sufficiently examined; furthermore, in the Japanese context, existing research on PRD is insufficient. Although PRD is thought to start with social comparison [10], the nature of social comparison differs relatively between individualistic cultures such as Western countries and collectivistic cultures such as East Asian ones [15]; thus, factors related to PRD may differ between Western and East Asian countries and need to be further explored. In East Asia, limited studies have investigated PRD, with the exception of the study by Kim et al. [16]. To understand the influence of culture on PRD, the present study also included an investigation of the relationship between PRD and cultural self-construal [17].

When health is compromised by increased PRD, the benefits to society as a whole are undermined due to changes such as increased health care costs [18]. In addition, the recent expansion of social media use may also aggravate feelings of PRD by increasing social comparisons [10] and causing negative effects such as low self-esteem [19]. Thus, the need to assess and monitor PRD at both social and personal levels is becoming increasingly important.

Subjective well-being focuses on people’s evaluation of their own lives and is considered to include both emotional and cognitive evaluations [20]. It is reportedly more important than money for many people, and scientific efforts are needed to create a better social environment [20]. Subjective well-being is now considered not only an indicator of mental health, but also an appropriate measure for social progress and a goal of public policy [21]. Data from 15 OECD countries have clearly demonstrated a decline in subjective well-being in the coronavirus disease (COVID-19) pandemic era, including indications that more than a quarter of the population has been at risk for depression and anxiety and increased feelings of loneliness, fragmentation, and disconnection from society [22]. Further, a report on the changes in subjective well-being by social class in Japan during the COVID-19 pandemic [23] suggests that the importance and scope of subjective well-being and PRD may be expanding, as it is also related to epidemiology. We believe it is essential to clarify the relationship between PRD and subjective well-being to determine possible interventions that consider related factors and to increase subjective well-being in the future. In addition, considering this topic in Japan, where subjective well-being is not high [24], this study may contribute not only to understanding the impact of cultural differences in PRD, but also to future social correction policy considerations in Japan.

### Research Aims

The main purpose of this study is to clarify the relationship between PRD and subjective well-being in Japan and to propose factors that may be related to them, which could, in turn, inform future measures to improve subjective well-being.

First, we demonstrated the external criterion-referenced validation of the J-PRDS5 [14] and investigated its relationship with subjective well-being. We examined the relationship between PRD and objective socioeconomic factors (such as age, sex, education level, and household income); personality factors (such as social comparison orientation and Big Five personality traits); and subjective socioeconomic factors (such as subjective socioeconomic status (SSS) and perceived social support [25]). Objective and subjective socioeconomic factors, Big Five personality traits, and social comparison orientation were included in the survey since their relationship with PRD has been well investigated [3,10,16], and they were considered suitable for evaluating the external criterion-referenced validity. Social support is shown to reduce life stress [26,27] and has been reported to buffer the relationship between depression and PRD [28]. Consequently, we hypothesize that perceived social support may mitigate feelings of PRD. The number of friends was also to be measured as an indicator of available support. Furthermore, as outcome measures for PRD, we first used indicators, such as health, stress, and materialism considered in existing studies [10,13,16], to confirm the replication of the association between PRD and these indicators.

To further examine the multidimensional relationship with subjective well-being, we used the various subjective well-being scales described in Section 2.4 (Measures) [29,30,31,32,33,34,35]. Although the Search for Meaning in Life [32] and Ideal Happiness Scale (IHS) [33,34,35] do not assess current happiness, they were included in the survey to examine the impact of PRD not only on current happiness, but also on the tendency to seek the meaning of life and the degree of happiness desired as the ideal happiness level.

Our second aim was to determine the extent to which PRD mediates the relationship of social comparison orientation with various well-being indices. We assumed this mediating relationship based on various existing studies, as follows. Regarding the relationship between social comparison orientation and subjective well-being, some data show that social comparison orientation is negatively correlated with various positive well-being indices [36]; studies have also demonstrated that people who make social comparisons more frequently experience more negative emotions [37]. In contrast, according to the downward comparison theory, recognizing oneself as superior to others contributes to increased happiness [38,39]. The assumption of a causal relationship between PRD and subjective well-being was based on the fact that relative deprivation based on income data has a negative impact on subjective well-being [7,11], and PRD is a predictor of self-rated health [13]. The association between social comparison and PRD has been well studied (e.g., [9,10,40]); social comparison orientation has already been analyzed as a precursor of PRD [9,10]. In addition, regarding the association between social comparison orientation and materialism, which is closely related to subjective well-being [41], PRD has already been shown to mediate some effects of social comparison orientation on materialism [10,16]. It is also possible that some or all of the effects of social comparison orientation on subjective well-being may be mediated by feelings of PRD. However, the association between social comparison orientation, PRD, and well-being has not been fully demonstrated.

Our third aim was to identify the factors that people with high PRD and high subjective well-being possess. One of our concerns was that some participants’ subjective well-being in the present study was maintained even when they had high PRD; we believe that analyzing the underlying factors behind this phenomenon might lead to future opportunities for maintaining and improving health and subjective well-being. Therefore, we compared the cases of individuals with high PRD and low subjective well-being with those with high PRD and high subjective well-being; we further clarified the differences in the varied factors of these two groups.

## 2. Materials and Methods

### 2.1. Ethics

Survey responses were solicited from individuals who had previously registered with the research firm, and participation was voluntary. Survey respondents could refuse to answer the questions, and the online questionnaire allowed them to discontinue or suspend their participation at any time. Anonymity of responses was guaranteed by the survey company, which was communicated to the respondents. It was also clearly stated in advance that privacy or any rights would not be violated and that anonymized or statistically processed data might be made public. Those who agreed to these conditions were asked to answer the questionnaire. The study was approved by the Ethics Committee of the Graduate School of System Design and Management, Keio University.

### 2.2. Participants and Procedures

This research is based on the same questionnaire survey as a previously published study on materialism [41]; however, the research topic differs. Participants were recruited through an online survey company (MyVoice Communications, Inc., https://www.myvoice.co.jp/ accessed on 12 December 2023), and a web-based questionnaire titled “Survey on Your Attitudes Toward Yourself and Society” was developed and administered. Participants were offered 70 Japanese yen (JPY; equivalent to 0.52 U.S. dollars as of August 2022) worth of reward points, which could be converted to Amazon gift cards or shopping coupons. To improve the quality of responses, two attention check items (e.g., “please select ‘do not agree at all’ for this question”) were included. To study the situation of people mainly in adulthood, in this research, the questionnaire survey was administered to adults aged 20–69 years. The survey ended upon its completion by 500 users—100 users each in their 20s, 30s, 40s, 50s, and 60s (50 men and 50 women). The users who gave inappropriate answers to the attention check items were removed by the online survey company to improve the accuracy of the analysis by eliminating the unreflective responses that can be found in Internet surveys. The target number of data entries was set at 500 persons because, even if the population is assumed to be infinite, a sample size of approximately 400 is considered sufficient, assuming a confidence coefficient of 95% and an acceptable margin of error of 5%. Israel [42] similarly suggested a sample size of 400 with a confidence level of 95% and a precision level of ±5%.

In preparing the dataset for the 500 participants, the online survey agency collected 668 participants’ data. A total of 168 participants’ data were excluded: 150 people who answered the attention check items incorrectly, 7 people who consistently chose the same option consecutively in three or more question groups (one question group means one scale group), and 11 randomly selected responses that were excluded by the survey company under the contract of providing data for 500 individuals. We, the researchers, were not involved in the exclusion process. The data of the final 500 participants were included in the analysis.

We first obtained participants’ demographic information: sex, age, region of residence, marital status, state of employment, educational level, annual household income, total number of family members living together, and total number of friends. To process the annual income option quantitatively, the middle value of each option was used, and for the highest open-ended option, the value based on the estimation equation by Parker and Fenwick [43] was used. We also used the middle value for each option for the number of friends, and for the highest open-ended option, we considered the number of friends as 35 for convenience. For the highest open-ended option concerning the number of family members, only 0.8% were present and were treated as six for convenience. Those who reported their education level as “other” were excluded from analyses regarding educational background. In the questionnaire, participants were asked to select the closest possible answer choice.

### 2.3. Participant Characteristics 

Participants’ sample characteristics are presented in Table 1.

### 2.4. Measures

The scales used are listed below in the order in which they were presented in the questionnaire. The Ideal Happiness question was asked between the 11-point subjective happiness question and the SHS questions.

#### 2.4.1. PRD

As a measure of the feelings of PRD, the Japanese version of the PRDS in Callan et al. [3,8]—the J-PRDS5 [14]—was used. The scale comprises five items, including “I feel deprived when I think about what I have, compared to what other people like me have.” Each item was rated on a six-point scale (1 = *strongly disagree* to 6 = *strongly agree*). The scores for the negative items were reversed, and the mean scores for each item were used to calculate the total score. The higher the score, the higher the feelings of PRD. Ohno et al. found that the J-PRDS5 had the same simple factor structure as the original version, with 57.0% variance explained, and reported that it was correlated with SSS, self-esteem, and general health as in existing studies [14]. In this study, the alpha coefficient indicating the reliability of the J-PRDS5 was 0.801 (see Section 3.3). The AVE (average variance extracted) for validity was 0.460, slightly below the Fornell–Larcker criterion [44], and the CR was 0.798. However, some studies (e.g., Huang et al. [45]) consider the Fornell–Larcker criterion of convergent validity to be met when the CR exceeds 0.6, even when the AVE is less than 0.5; thus, the J-PRDS5 score is treated in this study as having reliability and validity.

#### 2.4.2. Material Values

The Japanese version of the Material Value Scale (MVS) [46,47]—the J-MVS-P6 [41]—was used as the materialism scale. It comprises six items related to the three domains of materialism. As in the original version, respondents answered questions on a five-point scale. We calculated the material value scores as the mean of responses: the higher the score, the greater the tendency toward materialism.

#### 2.4.3. Subjective Happiness

To assess subjective well-being, we used two scales. First, we presented the question, “How happy are you at present?” [48]. To increase sensitivity toward the response, we asked respondents to answer the question (self-rated happiness) on an 11-point scale (0 = *very unhappy* to 10 = *very happy*). Second, we used the Japanese version [30] of Lyubomirsky and Lepper’s Subjective Happiness Scale (SHS) [29], which comprises four questions. Respondents answered the questions on a seven-point scale with one being the lowest and seven being the highest. The negative items were reverse-scored, and we calculated the total SHS score as the mean of four responses. A higher SHS score indicates a higher degree of happiness.

#### 2.4.4. Ideal Happiness

Takahashi et al. [33,34,35] developed an 11-point scale, with 0 indicating 100% unhappiness, 5 indicating half happiness and unhappiness, and 10 indicating 100% happiness. We referred to Takahashi et al.’s [33,34,35] idea that the difference between ideal (ideational) happiness and current happiness is important in assessing happiness. In this study, this ideal level of happiness is considered, and the difference between the ideal and current levels of happiness is calculated as a value indicating the gap between the ideal and reality on the 11-point self-rated happiness scale, using the single question item that was presented earlier.

#### 2.4.5. Meaning in Life

The Meaning in Life Scale (MLQ,), developed by Steger et al. [32], was employed to determine the strength of an individual’s sense of meaning in their life and their tendency to search for that meaning. The scoring characteristics of the Japanese version have been previously clarified [49]. The MLQ comprises the Search for Meaning in Life (MLQ Search) and the Presence of Meaning in Life (MLQ Presence) subscales. Overall, five questions from each subscale were used to derive the MLQ Search and MLQ Presence scores.

#### 2.4.6. Life Satisfaction

The Japanese version of the Satisfaction with Life Scale (SWLS) (1985), developed by Oishi [31], was used as an index of life satisfaction. The total score of five questions was calculated using a seven-point scale (minimum 5 points to maximum 35 points). A higher SWLS score indicates a higher degree of life satisfaction.

#### 2.4.7. General Health

A self-rated general health scale was used to assess participants’ general health. The following question was asked: “How do you feel about your current state of health?” [48]. Respondents answered the question on an 11-point scale (0 = *not at all healthy* to 10 = *very healthy*).

#### 2.4.8. Stress

To assess self-rated stress, we asked, “How much stress [e.g., because of hassles or demands] were you under recently?” [16,50]. Respondents answered the question on an 11-point scale (0 = *did not feel at all* to 10 = *felt very much*).

#### 2.4.9. SSS

For assessing SSS, we referred to Sato [51] and included the following question: “If you were to divide society as a whole into one to ten strata from the bottom to the top, which of these strata would you consider yourself to be in?” We provided 10 answer choices on a scale of 1 (lowest) to 10 (highest). The higher the value of this response, the higher the SSS.

#### 2.4.10. Social Comparison Orientation

The Japanese version of the Iowa Netherlands Comparison Orientation Measure (INCOM) by Gibbons and Buunk [36], which comprises two types of questions that measure ability comparison and opinion comparison orientation, was used to evaluate social comparison orientation. In Toyama’s Japanese version [52], the results of factor analysis showed that item 11 in the INCOM measured ability comparison, and item 9 in the INCOM had strong factor loadings for both ability and opinion comparison; thus, we eliminated item 9. The question items were surveyed on a five-point scale (1 = *not at all applicable* to 5 = *very applicable*). We calculated the mean scores of three items for opinion comparison and seven items for ability comparison; the scores were reversed for the negative items. The higher the mean score, the stronger the social comparison orientation in the ability or opinion comparison domains.

#### 2.4.11. Big Five

The Japanese version of the Ten Item Personality Inventory (TIPI-J) [53,54] was used as a scale to measure the five factors of Big Five personality. The questionnaire comprises two questions each on openness, conscientiousness, extraversion, agreeableness, and neuroticism (reversed scores of “emotional stability” item in the TIPI); the items are rated on a seven-point scale (1 = *do not agree at all* to 7 = *strongly agree*). Higher scores for a factor mean stronger personality tendencies corresponding to that factor.

#### 2.4.12. Independent/Interdependent Construal of Self

To investigate the influence of cultural self-construal on PRD, we included the Independent and Interdependent Construal of Self scale [17]. It comprises 10 questions on independent construal of self and 10 questions on interdependent construal of self. Questions include one item to determine one’s independent construal of self (e.g., “I always try to have my own opinions”) and one item to understand one’s interdependent construal of self (e.g., “I care what other people think of me”). Responses were obtained on a seven-point scale (7 = *completely true* to 1 = *not true at all*). Mean scores were calculated for both the independent and interdependent construals of self. Higher scores indicate stronger characteristics.

#### 2.4.13. Negative Affect

The Positive and Negative Affect Schedule (PANAS), translated into Japanese [55], was employed as a scale for rating negative moods. Participants answered questions on a six-point scale (1 = *not at all applicable* to 6 = *very applicable*). We randomized the order of the questions according to the guidelines.

#### 2.4.14. Perceived Social Support

To measure perceived social support, we used the Japanese version of the Multidimensional Scale of Perceived Social Support. The original scale was developed by Zimet et al. [25] and was adapted by Iwasa et al. [56]. All 12 questions were rated on a seven-point scale (1 = *do not agree at all* to 7 = *strongly agree*). The higher the score, the higher the perceived social support.

### 2.5. Analyses

R version 4.0.1 (v4.0.1; R Core Team 2020) was used for statistical processing. R is an open-source software environment and offers many functions for psychological analysis; it was used in this study because of its high convenience. For the first aim of the study, a correlation analysis between scale scores was conducted; for the second aim, a mediation analysis on social comparative orientation, PRD, and various subjective well-being scales was conducted using the lavaan package; and for the third aim, an analysis of differences in various indicators was conducted using groups, between the group with high PRD and high subjective well-being and the group with high PRD and low subjective well-being.

## 3. Results

### 3.1. Correlation Analysis for the Assessment of External Criterion-Referenced Validity of the J-PRDS5

The correlation coefficients with demographic factors and with various factors for evaluating the external criterion-referenced validity of the J-PRDS5 are shown in Table 2. PRD showed a significant negative correlation with age, household income, education level, and number of friends. No correlation was found between PRD and the number of family members. Sex was coded as 0 for male participants and 1 for female participants and showed a significant negative correlation with PRD. Results indicated that men tended to have a higher PRD than women. Marital status was coded as 1 for married and 0 for not married. Results revealed that PRD tended to be higher for those who were unmarried vs. those who were married. Work was coded as 0 for not working and 1 for working, and no correlation with PRD was found.

The correlation coefficients with household income were r = −0.32 in the US and r = −0.26 in the UK [13], with a slightly weaker but similar trend in Japan at r = −0.21. The association with education level was r = −0.15 in the US, but it did not have a significant r in the UK [13]. It was significant (r = −0.1) in Japan and was also similar to a previous study in the US. As in the previous study [14], a highly negative correlation with PRD existed for SSS. A weak negative correlation was found between PRD and age in Japan, similar to previous studies [9,14].

In the association between PRD and Big Five personality, significant negative correlations were found with openness, conscientiousness, extraversion, and agreeableness, and a significant positive correlation was found with neuroticism. A South Korean study reported [16] r = −0.25 for extraversion, r = −0.19 for conscientiousness, r = −0.30 for openness, and r = 0.31 for emotional stability (corresponding to neuroticism in this study with implications for reversal); for agreeableness, it was uncorrelated. Callan et al. [3] reported r = −0.33 for conscientiousness, r = −0.20 for openness, r = −0.32 for emotional stability, and r = −0.22 for agreeableness. A Japanese study noted r = −0.25 for extraversion, r = −0.11 for conscientiousness, r = −0.14 for openness, r = 0.38 for neuroticism, and r = −0.24 for agreeableness, which is approximately the same trend that has been previously found. 

In the association between PRD and social comparison orientation, a positive correlation was observed with ability comparison, with a correlation coefficient exceeding r = 0.4, while no correlation was found with opinion comparison. This was similar to the results in the US with r = 0.33 [9] and with the PRDS-3 in South Korea with r = 0.41–0.48 [16]. In contrast, for opinion comparison (INCOM opinion), no significant correlation with PRD was obtained, which was similar to the results of Callan et al. [9], which showed only slight significance or very low correlation coefficients. However, in the South Korean version [16], r = 0.19–0.29, a significant correlation was obtained with opinion comparison, and further exploration is needed to determine the factors behind this difference. A negative correlation was also found with perceived social support, with a correlation coefficient lower than −0.4. Concerning cultural self-construal, there was a weak negative correlation with the independent construal of self and a positive correlation with the interdependent construal of self.

### 3.2. Correlation Analysis for Assessment of the Relationships between PRD and Subjective Well-Being

Table 3 shows the correlation coefficients between PRD and the variables related to well-being and the alpha coefficients of each scale. The negative affect demonstrated by the PANAS showed a high correlation coefficient of r = 0.49 with PRD, whereas the correlation coefficient with PRD in the U.S. was r = 0.38 [13]. Concerning health, although the indices used in the studies were not necessarily identical to those in the current study, the correlation coefficients with global health and global physical health were r = −0.28 and −0.30 in the U.S., respectively, and r = −0.28 with global physical health in the UK [13]. The correlation coefficient with physical health was r = −0.22 in a South Korean study [16]; whereas, in Japan, the correlation coefficient with self-rated health was −0.30 (almost the same level as in the West). Concerning stress, the correlation coefficient was r = 0.34, compared to correlation coefficients of r = 0.54 [13] in the US and r = 0.28 in South Korea [16]. Concerning materialism, the correlation coefficients were r = 0.49 in the US, r = 0.44 in the UK [10], and r = 0.42 (MVS-9) and r = 0.33 (MVS-3) in South Korea [16]; whereas, it was r = 0.45 (J-MVS-P6) in Japan, indicating a similar correlation. 

The PRD was correlated with several happiness-related index scores in the predicted direction. In particular, highly negative correlations were found with the SWLS (r = −0.57), self-rated happiness (r = −0.63), and the SHS (r = −0.69). PRD was negatively correlated with MLQ Presence but not with MLQ Search. PRD and ideal happiness were also negatively correlated at r = −0.27, and the difference between ideal happiness and self-rated happiness were positively correlated at r = 0.4.

### 3.3. Mediation Analysis of PRD in the Relationship between Social Comparison Orientation and Subjective Well-Being Index Scores

The results of the mediation analysis of PRD for the association between social comparison orientation (ability comparison) and various well-being indices (including “materialism”) indicated that PRD significantly mediated (forming significant indirect effects) this relationship. The mediation analysis model is shown in Figure 1. Since social comparison orientation (opinion comparison) did not have a significant correlation with PRD, only social comparison orientation (ability comparison) was included in the mediation analysis model. Using the bootstrap method, 5000 re-samplings were performed, and each standardized coefficient between each variable was obtained to determine the direct (c’), indirect (a*b), and total (c) effects of social comparison orientation (ability comparison) on the various well-being index scores.

The analyzed well-being index scores, each standardized coefficient of each effect, each confidence interval, and variance accounted for (VAF: an index indicating the ratio of indirect effects to total effects) are summarized in Table 4. Although there is no consensus on the appropriate number of resamples for the bootstrap method, Preacher and Hayes [57] recommend at least 5000 resamples, which was followed in the current study. The analysis procedure, which was also employed by Zhao et al. [58], first involved checking whether the indirect effect (a*b) was significant, followed by the direct effect (c’). For VAFs, previous studies have employed a criterion of >0.2 for determining partial mediation and >0.8 for complete mediation [59]; the same criterion was used in this study. MLQ Search, which was not significantly correlated with PRD, was excluded from analyses.

Self-rated health, self-rated happiness, the SHS, the SWLS, and MLQ Presence, which indicate positive well-being, all showed high VAFs of 1 or more. All direct effects had significant positive standardized coefficients, while the total effect had a negative standardized coefficient and the indirect effect through PRD had a significant negative standardized coefficient. Regarding ideal happiness and the difference between ideal and self-rated happiness, the direct effect was not significant, and only the indirect effect via PRD was significant.

Negative affect (PANAS) and self-rated stress, which indicate a negative well-being state, showed significant positive values for both direct and indirect effects, and VAF exceeded 0.2, indicating that PRD partially mediated the variables. The VAF of 0.233 for materialism also replicated the partially mediated effect of PRD.

### 3.4. Analysis of the Related Factors That Maintain High Subjective Well-Being in High PRD

What are the related factors that maintain high subjective well-being even when PRD is high? Here, self-rated happiness is considered as the level of subjective well-being; we analyzed two groups of people whose subjective well-being remained above a certain level despite high PRD and whose subjective well-being was below a certain level: a group with median or above-median PRD and median or above-median self-rated happiness (GROUP1) and a group with median or above-median PRD and below-median self-rated happiness (GROUP2). We assessed the differences in the various variables including demographic factors, personalities, objective socioeconomic factors, and SSS between these two groups.

Wilcoxon rank sum test was applied to each variable between GROUP1 and GROUP2. Based on the *p*-values obtained from the test results, the Holm method was employed to check whether there were significant differences between the two groups (Table 5). A higher number of married individuals and those with higher SSS were found in GROUP1 compared to GROUP2. GROUP1 also tended to report higher extraversion, having significantly more friends, higher perceived social support, and a higher social comparison orientation (opinion comparison) than GROUP2.

## 4. Discussion 

### 4.1. External Criterion-Referenced Validation of the J-PRDS5 and Its Relation to Subjective Well-Being

The correlation analysis in this study confirmed the external criterion referenced-validity of the J-PRDS5 and found that PRD in Japan has a strong negative correlation with subjective well-being.

The relationship between feelings of PRD, measured by the J-PRDS5, and the diverse variables examined in this study, including objective socioeconomic factors (such as age, sex, education level, and household income), SSS, and personalities, was similar to that observed in previous studies using the original version of the PRDS [3,9,10,13]. The result that PRD was relatively strongly correlated with materialism and partially mediated the relationship between social comparison orientation (ability comparison) and materialism is also consistent with results in previous reports; it points to the similar nature of the J-PRDS5 in the original version [10] and the South Korean version [16]. These results further indicate the high external criterion-referenced validation of the J-PRDS5, which was lacking in the study by Ohno et al. [14]. This facilitated the evaluation of PRD in Japan. Although there may be room for further study on the full equivalence of the questionnaire items, we believe that this study allows for more reliable comparisons with PRD evaluation studies in other countries using the original version [3,8] or the Korean version [16]. 

The relationship of PRD to the human environment has not been well explored. In this study, PRD was not correlated with the number of family members but had a weak negative correlation with the number of friends and marital status and a relatively high negative correlation with perceived social support. The results suggest that a well-developed human environment, such as having numerous friends, being married, and feeling that one has social support, may be associated with low PRD. These results are consistent with a report suggesting an interaction between perceived social support and PRD that affects depression [28]. If PRD is considered to be a life stress, as Cobb [26] and Kawachi and Berkman [27] have shown, social support might have a mitigating effect on PRD or some positive effect on mental health and reduced PRD. Alternatively, those with a greater number of friends and higher perceived social support might be less susceptible to PRD. Further clarification of the mechanisms linking PRD and these factors related to the human environment is needed.

Thus far, the relationship between PRD and cultural self-construal has not been discussed. Notably, there was a positive correlation with the interdependent construal of self—which is typical in Asians—and a negative correlation with the independent construal of self—which is typical in Westerners. PRD might be more likely to increase in people who have a strong interdependent construal of self, as in Asians. This seems to have a high affinity with the suggestion that the need for general and upward comparison is stronger in East Asia [15], where collectivism is stronger. It is possible that the social comparison orientation that is characteristic of East Asia strongly induces PRD in Japan and, therefore, leads to subjective well-being that is not necessarily high [24] despite comparative affluence. Since this study suggested a link between cultural self-construal and PRD, further research is needed to translate the PRDS and evaluate PRD in other countries with different cultures, as well as to further investigate the strength of the association between PRD and subjective well-being. A study using the original version of the PRDS reported that people often imagined others who were more economically affluent than themselves as social comparators [40]. In addition, it was also reported that the people who came to mind while answering questions about PRD were more specific comparators in the order of friends, family members, and co-workers [13], but the comparators for experiencing PRD in Eastern cultures, including Japan, also need to be examined.

Although the relationship between PRD and health has been well investigated [13,16], the relationship with subjective well-being, which has various aspects, has not been fully explored. In our study, the PRD measured by the J-PRDS5 was significantly correlated with most of the well-being index scores. In particular, the relationship between happiness and PRD had not been previously shown. However, in our study, a highly negative correlation was found, with a correlation coefficient of nearly −0.7 with the SHS and a correlation coefficient of approximately r = −0.6 for the SWLS and self-rated happiness. Our finding suggests that PRD and subjective happiness are closely related beyond other indices, such as health. Our results also reveal negative correlations with ideal happiness. This might suggest that chronic negative emotions lead to a depressive state and lower expectations for future happiness. This would be consistent with Bjärehed et al.’s study [60], which found that when depressed individuals evaluated a positive event related to a personal goal, they believed that the event was unlikely to occur. Furthermore, the search for meaning in life was not correlated with PRD, and higher PRD might not increase the tendency to search for meaning in life. Contrastingly, the positive correlation coefficient for the presence of meaning in life might indicate that high PRD tended to make people less likely to believe that life had meaning. Further clarification of the mechanisms that produce this trend is also needed.

### 4.2. PRD Mediation between Social Comparison Orientation (Ability Comparison) and Subjective Well-Being

The mediation analysis in this study confirmed a positive direct effect of social comparison orientation (ability comparison) on many subjective well-being indicators and a significant negative indirect effect via PRD.

PRD in Japan played a mediating relationship between social comparison orientation (ability comparison) and materialism and various well-being index scores. Although it is known that PRD mediates the relationship between materialism and ability comparison [10,16], this is the first report in which PRD has been shown to mediate the relationship between scores on various subjective well-being indices and ability comparison.

PRD on the associations between social comparison orientation (ability comparison) and subjective well-being index scores, such as subjective happiness, life satisfaction, self-rated health, and MLQ Presence, was fully mediated (VAF exceeded one). The negative effects of increased social comparison orientation (ability comparison) on subjective happiness, life satisfaction, self-rated health, and MLQ Presence seemed to be mostly due to PRD. The direct effects of social comparison orientation (ability comparison) on subjective happiness, life satisfaction, self-rated health, and MLQ Presence, excluding the mediation effect, were positive. Some data have shown that social comparison orientation is negatively correlated with various positive well-being indices [36] and that people who make social comparisons more frequently have been thought to experience more negative emotions [37], which may be largely because of PRD. The present results also suggest that in addition to social comparison orientation (ability comparison) having a downward effect on subjective well-being via PRD, it may conversely enhance subjective well-being index scores. This may be consistent with the downward comparison theory [38,39], which suggests that self-esteem and happiness are enhanced by a strong tendency to consider oneself as superior to others.

Negative affect (PANAS) and self-rated stress were partially mediated by PRD in the same way as materialism, indicating that part of the influence of social comparison orientation (ability comparison) on both factors may be generated through PRD. 

Reportedly, social media is a place where social comparisons are made at a high frequency; individuals who have a high social comparison orientation may be negatively affected [19]. One could argue that this may be due to a heightened feeling of PRD. However, the present results and the downward comparison theory suggest that social media may also have positive effects outside of the feelings of PRD. In the future, applied research should be conducted to clarify the kind of social media usage that can suppress the induction of PRD and bring out the positive aspects of social comparison.

### 4.3. Factors Related to High PRD and High Subjective Happiness in People

Group analyses in this study revealed several factors, especially those related to the human environment, that may be associated with higher subjective well-being even when PRD is high.

This study explored the factors that are related to high subjective happiness even with high PRD. Participants with median or above-median PRD were divided into groups according to their subjective happiness, and the differences were examined. Findings revealed that the group with higher PRD and higher subjective happiness tended to be married and reported higher SSS and extraversion. The group with higher PRD and higher subjective happiness also tended to have significantly more friends, higher perceived social support, and a higher social comparison orientation (opinion comparison). Having more friends, tending to compare opinions more, and receiving more social support might be related to maintaining high subjective happiness regardless of PRD. Perceived social support and the number of friends were also negatively correlated with PRD, as presented in Section 3.1 in Results, and might be important factors in both not increasing PRD and maintaining high subjective happiness even when PRD is high.

Although the feelings of PRD are directly disadvantageous to those who experience them, negative effects on health and well-being may be relevant to the larger society in the future, as with addressing increasing healthcare costs and gaps in social policies [18]. As a basic premise for improving ill health and mitigating unhappiness, which are negative consequences of PRD, the first step in considering medical and social policies would be to monitor PRD and consider measures to deal with it based on the monitored results. Although this study was limited to an exploratory investigation of factors associated with high subjective well-being despite high PRD, it is necessary to further clarify the mechanism by which factors such as the number of friends, social support, and comparison of opinions are associated with the maintenance of high subjective well-being while having high PRD. If the mechanism is clarified, it may lead to consideration of policies to maintain and enhance subjective well-being by improving the human environment, such as with social support, as a measure to cope with PRD. A recent study which found that hope mitigates PRD and reduces risk-taking behavior [61] is an example of research that should be consulted, and the results are interesting in showing coping strategies that could be incorporated in practice.

### 4.4. Limitations

Since this study is an analysis based entirely on self-reported responses and does not include objective observations, it is prone to distinct biases. Notably, this study was conducted through a web survey, and it is highly likely that people with low information literacy are not included. Therefore, these trends may not necessarily be generalizable to the Japanese population at large. In the future, it will be necessary to conduct surveys using methods other than web surveys.

Regarding the alpha coefficient of the TIPI-J for the Big Five assessment used in this study, it was lower than the general standard; it is desirable to use a questionnaire that is not a shortened version in the future.

We focused on perceived social support and the number of friends, which are factors that can alleviate life stress. However, factors that do not increase PRD and factors that maintain subjective well-being should be examined more extensively in the future.

Causal inferences cannot be made in terms of our third aim. In this study, the number of friends, perceived social support, and opinion comparison were extracted as factors that contribute to higher subjective well-being even when PRD increases; however, further empirical verification is necessary to determine the effectiveness of these factors.

Additionally, in this study, we did not analyze how sex differences affected these results. It is also necessary to investigate the situation of older adults aged 70 years and above, who were excluded from this study.

## 5. Conclusions

From our evaluation of the relationship between PRD and subjective well-being, by considering various related aspects and in the context of Japan, some notable findings emerged. First, this study demonstrated that the J-PRDS5, which had been tentatively developed, had external criterion referenced-validity, allowing for comparisons with PRD studies in other countries. Second, we showed, for the first time, that PRD and subjective well-being, which have various complex aspects, are closely related in the Japanese context and that when including social comparison orientation, a structural relationship exists between the three factors. PRD is a full mediator between ability comparison and subjective well-being, and ability comparison has a positive effect on subjective well-being except for the effect of PRD. Third, we identified factors, including a well-developed human environment, that may be involved in cases where subjective well-being is maintained above a certain level even when PRD is high. This may be useful for future studies on measures to improve subjective well-being. It is hoped that further translations of the PRDS and evaluations of PRD in other countries will be conducted to clarify the details of the mechanism of the relationship between PRD and subjective well-being and to promote the consideration of measures to improve subjective well-being.

## Figures and Tables

**Figure 1 behavsci-13-00158-f001:**
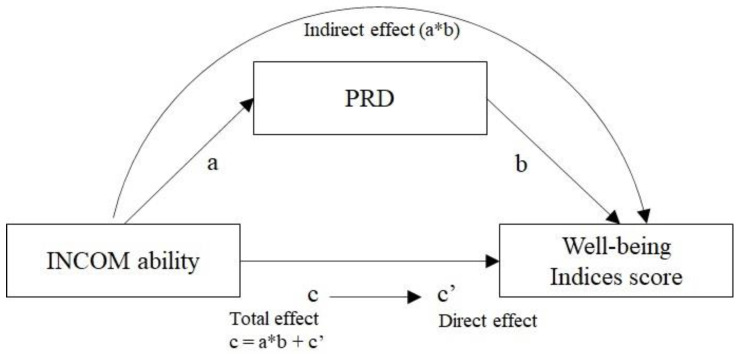
Mediation analysis model.

**Table 1 behavsci-13-00158-t001:** Sample characteristics.

*N*	500	Education (%)		Work Status (%)	
*M* age *(SD)*	44.83 (14.105)	Junior high school graduation [item 1]	1.6	Working	70.6
**Number of family members living together (%)**		High school graduation [item 2]	28.4	Not working	29.4
Living alone	15.2	Vocational school, junior college and technical college graduation [items 3 & 4]	22.4	**Income (%)**	
Living with 1	35.4	University graduation [item 5]	42	<2,000,000 JPY	9.4
Living with 2	24	Postgraduate degree [item 6]	5	>=2,000,000 JPY; <3,000,000 JPY	8.2
Living with 3	16.6	Other [item 7]	0.6	>=3,000,000 JPY; <4,000,000 JPY	13.4
Living with 4	6.8	**Number of friends (%)**		>=4,000,000 JPY; <6,000,000 JPY	17.2
Living with 5	1.2	None	12.4	>=6,000,000 JPY; <8,000,000 JPY	16.2
Living with >=6	0.8	1–5 persons	44.2	>=8,000,000 JPY; <10,000,000 JPY	10
**Marital status (%)**		6–10 persons	23.8	>=10,000,000 JPY	10.6
Married	46.2	11–20 persons	9	I don’t know	15
Not married	53.8	21–30 persons	3.4		
		>31 persons	7.2		

JPY: Japanese Yen.

**Table 2 behavsci-13-00158-t002:** Correlation between PRD and demographic factors and variables for the assessment of external criterion-referenced validity of the J-PRDS5.

	N	Mean	SD	Cronbach’s α	Correlation with PRD (PRDS)	*p*
Age (years)	500	44.828	14.105	-	−0.166	***
Sex ^†^	500	0.500	0.501	-	−0.112	*
Marital status ^‡^	500	0.538	0.499	-	−0.194	***
Number of family members	500	2.712	1.247	-	−0.011	
Education	497	3.205	0.966	-	−0.095	*
Work status ^§^	500	0.706	0.456	-	0.019	
Income (million Japanese yen)	425	5.880	3.440	-	−0.210	***
Number of friends	500	7.950	9.267	-	−0.210	***
Big 5: Openness	500	7.466	2.416	0.445	−0.138	**
Big 5: Conscientiousness	500	7.728	2.530	0.561	−0.110	*
Big 5: Extraversion	500	7.098	2.723	0.598	−0.248	***
Big 5: Agreeableness	500	9.516	2.197	0.342	−0.240	***
Big 5: Neuroticism	500	8.648	2.412	0.481	0.383	***
Social comparison orientation of ability (INCOM ability)	500	2.708	0.853	0.890	0.410	***
Social comparison orientation of opinion (INCOM opinion)	500	2.959	0.929	0.793	0.014	
Interdependent	500	4.379	0.846	0.838	0.285	***
Independent	500	4.409	0.857	0.849	−0.156	***
Subjective socioeconomic status	500	4.988	1.917	-	−0.544	***
Social support	500	4.410	1.359	0.952	−0.410	***

* *p* < 0.05, ** *p* < 0.01, *** *p* < 0.001. ^†^ Male and female participants were 50% each, and this result indicates a trend toward a higher PRD in male participants. ^‡^ Married participants comprised 46.2%. of the sample. This result indicates that PRD tended to be higher among the unmarried. ^§^ The percentage of those working was 70.6%.

**Table 3 behavsci-13-00158-t003:** Correlation between PRD and variables related to well-being.

	*Mean*	*SD*	1		2		3		4		5		6		7		8		9		10		11		12
1. PRD (J-PRDS5)	3.081	0.951	(0.801)																						
2. Negative affect (PANAS)	22.458	9.215	0.490	***	(0.935)																				
3. Self-rated stress	6.142	2.494	0.339	***	0.453	***	-																		
4. Self-rated health	5.944	2.444	−0.303	***	−0.360	***	−0.355	***	-																
5. Materialism (J-MVS-P6)	2.800	0.767	0.451	***	0.325	***	0.230	***	−0.033		(0.776)														
6. Self-rated happiness	5.988	2.368	−0.627	***	−0.480	***	−0.391	***	0.462	***	−0.293	***	-												
7. Subjective happiness (SHS)	4.373	1.195	−0.691	***	−0.532	***	−0.423	***	0.446	***	−0.344	***	0.822	***	(0.843)										
8. Life satisfaction (SWLS)	18.370	6.575	−0.570	***	−0.378	***	−0.400	***	0.348	***	−0.300	***	0.711	***	0.749	***	(0.899)								
9. MLQ Presence	18.778	6.744	−0.326	***	−0.229	***	−0.214	***	0.239	***	−0.111	*	0.434	***	0.539	***	0.589	***	(0.896)						
10. MLQ Search	21.746	6.137	0.062		0.157	***	0.059		0.090	*	0.230	***	0.076		0.089	*	0.122	**	0.457	***	(0.899)				
11. Ideal happiness	6.670	1.949	−0.269	***	−0.211	***	−0.145	**	0.289	***	−0.059		0.470	***	0.353	***	0.256	***	0.149	***	0.093	*	-		
12. Difference between ideal happiness and self-rated happiness	0.682	2.251	0.427	**	0.322	***	0.286	***	−0.235	***	0.257	***	−0.645	***	−0.559	***	−0.526	***	−0.327	***	0.001		0.371	***	-

* *p* < 0.05, ** *p* < 0.01, *** *p* < 0.001. Cronbach’s α coefficient in parentheses. PRD: personal relative deprivation, PANAS: Positive and Negative Affect Schedule, J-PRDS5: Japanese translation of the Personal Relative Deprivation Scale, J-MVS-P6: Japanese version of the Material Values Scale, SHS: Subjective Happiness Scale, SWLS: Satisfaction with Life Scale, MLQ Presence: Meaning in Life Questionnaire (presence of meaning in life), MLQ Search: Meaning in Life Questionnaire (search for meaning in life).

**Table 4 behavsci-13-00158-t004:** Results of the mediation analysis.

Well-Being Indices	(Standardized) a		(Standardized) b		Indirect Effect (Standardized) a*b		Direct Effect: (Standardized) c’		Total Effect: (Standardized) c		VAF
Negative affect (PANAS)	0.41 [0.301, 0.514].	*****	0.396 [0.292, 0.493].	*	0.162 [0.109, 0.223]	*	0.231 [0.134, 0.322]	*	0.393 [0.296, 0.483].	*	0.412
Self-rated stress	0.41 [0.301, 0.514].	*	0.285 [0.187, 0.376]	*	0.117 [0.074, 0.17].	*	0.132 [0.038, 0.227]	*	0.249 [0.167, 0.345]	*	0.470
Self-rated health	0.41 [0.301, 0.514].	*	−0.351 [−0.452, −0.248]	*	−0.144 [−0.207, −0.093]	*	0.117 [0.017, 0.216]	*	−0.027 [−0.127, 0.065]		5.333
Materialism (J-MVS-P6)	0.41 [0.301, 0.514].	*	0.291 [0.199, 0.378]	*	0.119 [0.078, 0.17].	*	0.391 [0.3, 0.487]	*	0.51 [0.426, 0.596].	*	0.233
Self-rated happiness	0.41 [0.301, 0.514].	*	−0.675 [−0.746, −0.599]	*	−0.277 [−0.356, −0.197]	*	0.116 [0.038, 0.199]	*	−0.161 [−0.259, −0.057]	*	1.720
Subjective happiness (SHS)	0.41 [0.301, 0.514].	*	−0.725 [−0.791, −0.657]	*	−0.297 [−0.378, −0.217]	*	0.083 [0.009, 0.156]	*	−0.215 [−0.317, −0.112]	*	1.381
Life satisfaction (SWLS)	0.41 [0.301, 0.514].	*	−0.612 [−0.68, −0.545]	*	−0.251 [−0.325, −0.18]	*	0.102 [0.012, 0.191]	*	−0.148 [−0.253, −0.041]	*	1.696
MLQ Presence	0.41 [0.301, 0.514].	*	−0.375 [−0.478, −0.269]	*	−0.154 [−0.222, −0.097]	*	0.118 [0.003, 0.224]	*	−0.035 [−0.14, 0.07]		4.400
Ideal happiness	0.41 [0.301, 0.514].	*	−0.295 [−0.412, −0.184]	*	−0.121 [−0.184, −0.072]	*	0.063 [−0.033, 0.163]		−0.058 [−0.15, 0.04]		2.086
Difference between ideal happiness and self-rated happiness	0.41 [0.301, 0.514].	*	0.455 [0.349, 0.563]	*	0.186 [0.126, 0.263]	*	−0.068 [−0.163, 0.032]		0.118 [0.018, 0.224]	*	1.576

a, b, c and c’ are pass coefficients in the mediation analysis model in the Figure 1. Comma-separated numbers in parentheses indicate 95% confidence intervals. * *p* < 0.05. PANAS: Positive and Negative Affect Schedule, J-MVS-P6: Japanese version of the Material Values Scale, SHS: Subjective Happiness Scale, SWLS: Satisfaction with Life Scale, MLQ Presence: Meaning in Life Questionnaire (presence of meaning in life).

**Table 5 behavsci-13-00158-t005:** Comparison of variables between the group with high PRD and high subjective well-being (GROUP1) and the group with high PRD and low subjective well-being (GROUP2) †.

	High PRD and High SWB (GROUP1; n = 104)			High PRD and Low SWB (GROUP2; n = 201)			Wilcoxon Rank Sum Test		
Variables	*Median*	*Mean*	*SD*	*Median*	*Mean*	*SD*	*p*		*Judgment by Holm’s Method*
Age (years)	44.00	44.78	14.40	41.00	42.78	13.28	n.s.	2.69 × 10^−1^	
Sex	1.00	0.54	0.50	0.00	0.40	0.49	<0.05	2.44 × 10^−2^	Decision on hold
Marital status	1.00	0.66	0.47	0.00	0.39	0.49	<0.001	7.77 × 10^−6^	*
Number of family members	3.00	2.89	1.25	2.00	2.58	1.21	<0.05	3.17 × 10^−2^	Decision on hold
Education level	4.00	3.37	0.88	3.00	3.09	1.05	<0.05	2.33 × 10^−2^	Decision on hold
Work status	1.00	0.76	0.43	1.00	0.70	0.46	n.s.	2.84 × 10^−1^	
Income (million Japanese yen)	5.00	6.04	3.50	5.00	5.04	3.37	<0.05	1.51 × 10^−2^	Decision on hold
Number of friends	8.00	9.93	10.04	3.00	6.48	8.99	<0.001	3.16 × 10^−6^	*
Big 5: Openness	7.00	7.49	2.35	7.00	7.07	2.26	n.s.	2.06 × 10^−1^	
Big 5: Conscientiousness	8.00	8.02	2.27	8.00	7.29	2.63	<0.05	1.58 × 10^−2^	Decision on hold
Big 5: Extraversion	8.00	7.76	2.53	6.00	6.22	2.47	<0.001	8.71 × 10^−7^	*
Big 5: Agreeableness	9.00	9.43	2.30	9.00	8.94	2.06	<0.05	4.75 × 10^−2^	Decision on hold
Big 5: Neuroticism	9.00	8.90	2.34	9.00	9.42	2.24	n.s.	1.29 × 10^−1^	
INCOM (ability)	2.86	2.88	0.77	3.00	2.92	0.81	n.s.	5.74 × 10^−1^	
INCOM (opinion)	3.33	3.22	0.85	3.00	2.82	0.88	<0.001	1.74 × 10^−4^	*
Independent	4.30	4.38	0.83	4.20	4.24	0.80	n.s.	1.87 × 10^−1^	
Interdependent	4.35	4.52	0.75	4.40	4.48	0.84	n.s.	7.30 × 10^−1^	
Subjective socioeconomic status	6.00	5.50	1.52	4.00	3.75	1.61	0.001	2.69 × 10^−16^	*
Social support	4.83	4.82	1.00	3.92	3.72	1.33	<0.001	1.36 × 10^−12^	*

* *p* < 0.05; n.s.: not significant. † The total for GROUP1 and GROUP2 is 305; this is because 195 participants with low PRD (less than the median) were excluded from the analysis.

## Data Availability

Data can be provided by the corresponding author with reasonable justification.

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
