# Peer review of "Feelings of Personal Relative Deprivation and Subjective Well-Being in Japan"

_behavsci, 2023, doi:10.3390/bs13020158_

Round 1
Reviewer 1 Report
1.Abstract: it could be improved by added future important of research results.
2.Introduction: to be introduced some relevant reference betweend COVID-19 period. To clarify the semnification of wellbeing concept, used in this article.
Some example:
OECD (2021), COVID-19 and Well-being: Life in the Pandemic, OECD Publishing, Paris, https://doi.org/10.1787/1e1ecb53-en.
CRISTINA, T. Ţ., & LUANA, M. M. (2017). THE WELLBEING OF SMALL CHILDREN AT THE BEGINNING OF THE PRIMARY SCHOOLING. Romanian Journal of Experimental Applied Psychology, 8.
OECD (2020), How's Life? 2020: Measuring Well-being, OECD Publishing, Paris, https://doi.org/10.1787/9870c393-en.
3.I would be important to add respondents ages, related to the criteria analyzed in results.
4.About research population and instrument: was a pretest moment?
5.What are the limits of research? What are the real contribution to the filed domain?
6.How about the results? How about the main contribution and originality?
Reviewer 2 Report
I suggest shortening the introduction. Please focus on the key issues
In the section: Research Aims, I propose to present only the goals - the main goal and specific goals. It seems to me that referring to examples from the literature is redundant.
The paragraph: Participants and Procedures proposes to be divided into: Procedures and Characteristics of the study group
I have the impression that the discussion is mainly a discussion of the results of my own research. It proposes to add literature related to the problem from 2022-2018. Compare your research results with other authors.
In general, the article is too long, it is proposed to shorten it and add current literature on the subject.
Reviewer 3 Report
This paper approaches a very interesting, actual and emergent issue, being this research a good contribution to the field in which it is located, namely wellbeing and health.
The text has a good structured and is organized according to the standards of the academic papers. The language is also good and no spelling errors were found. A robust study design and data analysis was carried out conferring to the article a significant quality.
Some sentences and the coherence between some sections revealed some weaknesses that claim for authors attention. Details are described by section as follows:
Abstract doesn’t refer to the sample size of the study and it is highly recommended.
Introduction reveals appropriate and the aims of the study are very detailed and theoretical anchored, more than it is usual. To synthesize the three aims of the study could become the research easier to understand, only a suggestion.
Material and Methods: For sample characterization or sociodemographic a small table could elucidate better than the presented description in the text, which becomes a little confuse given the amount of data. Several scales (14) are presented in detail and in the final, about analyses, only the software R is mentioned. As a recommendation, the type of statistical analyses should be described in this section.
Results: Good results are obtained and well presented in tables and described in the text. Nevertheless, results could be better organized and more understandable if presenting exclusively the results of this study in this section and move the comparisons with other study (literature) for discussion. In table 1 some results do not make sense in the presented form, namely mean and standard deviation for sex and marriage, because this are nominal and no metric variables. Frequencies and/or percentage for each category is more appropriate. Also, table 4 present results of two groups, which totalize 305 participants. Given the sample size of 500, some explanation is missing in order to understand what happened with the others.
Discussion can be improved with a better synthesis of the global results obtained in the first paragraph. After, consider to move the comparisons of other countries results mentioned in results section, to discussion, as recommended above. In addition, some declarations in this section seems contradictory with the previous, namely comparing what is said in results and discussion, more precisely lines 366-367 and 370-371 with what is referred in lines 582-583 and 587-588. More, these two last sentences (lines 582-583 and 587-588) seems contradictory between themselves. Good attention is recommended in order to clarify these points.
Limitations are mentioned that is clearly appropriate, but justification for the limitations do not need to be so long.
Round 2
Reviewer 3 Report
Authors attended the majority of the recommendations and provide good justifications when they want to maintain as it was in the first version.
An good improvement of the paper was done and now it is easier to understand and read.